# Quality Management System for an IoT Meteorological Sensor Network—Application to Smart Seoul Data of Things (S-DoT)

**DOI:** 10.3390/s23052384

**Published:** 2023-02-21

**Authors:** Moon-Soo Park, Kitae Baek

**Affiliations:** 1Department of Climate and Environment, Sejong University, Seoul 05006, Republic of Korea; 2Climate Change & Environmental Research Center, Sejong University, Seoul 05006, Republic of Korea

**Keywords:** internet of things, quality control, Smart Seoul Data of Things, quality management system for S-DoT meteorological sensor network, heatwave

## Abstract

Meteorological data with a high horizontal resolution are essential for user-specific weather application services, such as flash floods, heat waves, strong winds, and road ice, in urban areas. National meteorological observation networks, such as the Automated Synoptic Observing System (ASOS) and Automated Weather System (AWS), provide accurate but low horizontal resolution data to address urban-scale weather phenomena. Many megacities are constructing their own Internet of Things (IoT) sensor networks to overcome this limitation. This study investigated the status of the smart Seoul data of things (S-DoT) network and the spatial distribution of temperature on heatwave and coldwave event days. The temperature at above 90% of S-DoT stations was higher than that at the ASOS station, mainly because of different surface covers and surrounding local climate zones. A quality management system for an S-DoT meteorological sensor network (QMS-SDM) comprising pre-processing, basic quality control, extended quality control, and data reconstruction using spatial gap-filling was developed. The upper threshold temperatures for the climate range test were set higher than those adopted by the ASOS. A 10-digit flag for each data point was defined to discriminate between normal, doubtful, and erroneous data. Missing data at a single station were imputed using the Stineman method, and the data with spatial outliers were filled with values at three stations within 2 km. Using QMS-SDM, irregular and diverse data formats were changed to regular and unit-format data. QMS-SDM application increased the amount of available data by 20–30%, and significantly improved data availability for urban meteorological information services.

## 1. Introduction

High-quality meteorological data with a high horizontal resolution in urban areas are essential for user-specific application services for various weather phenomena, such as flash floods, heat waves, strong winds, drought, and road ice [1,2,3,4,5]. Urban building blocks have a 10 m horizontal scale and 1 min temporal scale inhomogeneity of surface temperature [2,6]. High-resolution and low-cost sensor networks are being used in many megacities worldwide to compensate for the low horizontal resolution of nationwide meteorological observation network [3]. These networks include meteorological sensors as well as air quality, noise, and vibration sensors to deliver useful information to the citizen. The “Array of Things” project developed in Chicago is an example [4].

National meteorological and air quality data have been maintained and operated by national meteorological administrations (e.g., the National Weather Service in the U.S. and Korea Meteorological Administration in Korea) and national environmental administrations (e.g., the Environmental Protection Agency in the U.S. and Ministry of Environment in Korea), respectively. The automated synoptic observing system (ASOS) and automatic weather system (AWS), controlled by the World Meteorological Organization (WMO), are superior in data quality, and quality control is well organized [7]. Automatic sampled and transmitted data have errors not only in the sensor itself but also in electric connections and telecommunication [5]. Sensor errors fall into the following five categories: (1) errors due to the failure of some system component, (2) calibration errors, (3) sensor drift in gain or bias-type errors, (4) exposure errors, and (5) system noise [8]. Electric and telecommunication errors can be random or systematic.

Quality management provides the principles and methodological framework for the operation and coordinates activities regarding quality [7]. Quality control is associated with components that are used to ensure that the quality requirements are met, and it includes all operational techniques and activities used to fulfill the quality requirements. Quality management aims to ensure that data meet the requirements for uncertainty, resolution, continuity, homogeneity, representativeness, timeliness, and format for the intended application at a minimum practical cost.

Most meteorological observation networks have their own quality control systems. For example, Nordic countries have adapted station QC (QC0), real-time QC (QC1), non-real-time QC (QC), and human QC (HQC) [9]. Oklahoma mesonet has adapted climate range, step, persistence, spatial, and like-instrument tests [10,11]. Moreover, the Korea Meteorological Administration (KMA) developed a quality management system (QMS) for meteorological data [12]. All meteorological data were collected into the Combined Meteorological Information System (COMIS), and their quality was assessed and controlled using a real-time quality control system for meteorological observation data (RQMOD) developed by the KMA, which consists of physical limit, climate range, step test, persistence check, internal consistency, and median filter [13].

Recently, many large cities have been increasing their Internet of Things (IoT) sensor networks to achieve higher horizontal resolution. Seoul, South Korea, established and expanded an IoT sensor network, called the Smart Seoul Data of Things (S-DoT) [2]. The S-DoT aimed not only to assist municipal policies related to living quality, such as air quality, noise, and odor but also to deliver useful information including meteorological variables to their citizens. Most IoT networks use micro-electro-mechanical system (MEMS) sensors, which are much cheaper than the high-quality and expensive sensors used by national meteorological observation networks. Additionally, most sensors are installed over unsuitable observational environments between urban obstacles, such as buildings. Thus, IoT sensor networks require a more rigorous QMS than conventional meteorological observation networks. Zhang et al. addressed the importance of data quality in IoT networks [14]. Missing and incorrect values in these networks should be removed or replaced with correct values based on quality control [15].

This study aimed to develop a QMS for an IoT meteorological sensor installed in a non-ideal environment and apply it to S-DoT meteorological sensors in Seoul, South Korea. The air temperatures obtained by the S-DoT were compared with those obtained by the ASOS and AWS operated by the KMA on heatwave and coldwave event days. The QMS for S-DoT meteorological sensors (QMS-SDM) was applied to the S-DoT data from August 2020 to July 2021. This study highlights the application of QMS-SDM.

## 2. Seoul Data of Things (S-DoT)

### 2.1. S-DoT

The S-DoT network was installed in Seoul City, spanning over an area of 605 km^2^ in order to not only assist municipal policies related to living quality such as air quality, vibration, and odors but also deliver useful information including meteorological variables to the Seoul citizens. In total, 850 and 220 stations were installed in 2019 and 220, respectively [2]. The network had a horizontal resolution of 0.75 km, which was much higher than that of the AWS (4.5 km) by the KMA. There was only one ASOS station in Seoul City (Figure 1b). Reliable meteorological measurements in Seoul City were confined to the 1 ASOS and 25 AWS stations operated by KMA. However, more than half of these were installed over a rooftop on a building, and the remaining ones were installed over suburban and not realistic urban surfaces.

The sensors were embedded in an enclosure (250 mm (H) × 150 mm (W) × 350 mm (H)), and the stations were selected for fulfilling the gridded data (75%), air quality (21%), policy needs (3%), and ventilation paths (1%).

Most S-DoT stations were installed over realistic urban areas surrounded by middle- or high-rise compact buildings in downtown areas (87%), while some were installed near mountains or parks (4%) and riversides (9%) (Table 1). Most downtown stations were located in residential, roadside, or commercial areas. Enclosures were mostly installed at 3–4 m on streetlight polls, closed-circuit television polls, or smart polls, while some were installed on the wall of a building directed to the east, south, west, or on a rooftop.

### 2.2. Data Characteristics

Basic stations (Type A) measure temperature, relative humidity, noise, illumination level, ultraviolet, 3-directional vibration, PM_10_ (particulate matter with diameter ≤10 μm), and PM_2.5_ (particulate matter with diameter ≤2.5 μm) concentrations. Some stations measure wind speed and direction, or a globe temperature in addition to the parameters measured by the basic station (Type B). Moreover, some stations measure the concentration of air pollutants, such as CO, NO_2_, and SO_2_ (Type C), while others measure NH_3_ and H_2_S concentrations for odor detection (Type D). In this study, more than 93% (992) of all stations were of the basic type (Type A); additionally, 32, 20, and 20 stations were of Type B, C, and D, respectively (Table 2). In total, 1,064 temperature and relative humidity sensors, and 32 wind speed and direction sensors were installed. Wind speed and direction sensors were not distributed evenly but were installed along the river to monitor the urban ventilation paths or wind flows.

The S-DoT data were available in the format of a comma-separated value (.csv), json, or open Application Programming Interface format. Notably, the data format was changed several times. S-DoT was designed to sample data every 2 min, but there were many missing data due to electrical or telecommunication problems.

## 3. Climatology on Heatwave and Coldwave Event Days

The threshold values for climatic range tests should be determined for quality control. The climatology of heatwave and coldwave event days was investigated. Heatwave and coldwave events are one of the synoptic-scale weather phenomena. However, urban areas tend to show higher temperatures during a heatwave period due to extra heat storage in and release from materials such as concrete and asphalt [6,16]. The spatiotemporal distributions of temperature obtained using the S-DoT IoT sensors were compared with those obtained from KMA-operated ASOS.

### 3.1. Heatwave Event Day (24 July 2021)

A heatwave event day was selected as 24 July 2021, which was the hottest day in Seoul in 2021. The average, maximum, and minimum temperatures in the Seoul ASOS station were recorded as 31.7, 36.5, and 26.7 °C, respectively.

Figure 2 shows the horizontal distribution of the daily mean, maximum, and minimum temperatures on the heatwave event day. The spatially averaged daily mean temperature was 33.1 °C, which was higher than that recorded by the ASOS by 1.4 °C (Figure 2a). Further, 89% of the S-DoT stations showed a higher daily mean temperature than the ASOS, whereas only 11% showed lower daily mean values. The spatially averaged maximum temperature was 39.3 °C, which was higher by 2.8 °C than that recorded by the Seoul ASOS (Figure 2b). Moreover, 97% of the S-DoTs exhibited a higher maximum temperature than that recorded by the ASOS. The spatial average of the daily minimum temperature was 28.3 °C, which was higher than ASOS temperature by 1.6 °C. Notably, the temperatures in central Seoul areas were much higher than those in the northern, southernmost, and eastern areas.

The highest value of the daily maximum temperature was recorded at the Doksan Library Station, which is located near mountainous regions, although temperatures near mountainous regions were generally lower than those in the surrounding areas (Figure 3c). The second- and third-highest daily maximum temperatures were recorded at the Changsin and Myeongryun Stations in Jongro-gu District (Figure 2b and Figure 3a,b). The sensor box of the above stations was attached to the west-facing or south-facing wall (Figure 3). Thus, the sensors installed on the wall could not represent the surrounding areas. Solar radiation heats up the wall directly, and the heated wall could affect the sensor through conduction [6,16].

The stations can be clustered according to temporal variation patterns. In this study, the time series of all stations were classified into 5 clusters using the dynamic time warping (DTW) clustering technique (Figure 4) [17,18]. Figure 5 shows the time series of centroid temperatures for each cluster. Cluster 1, which had stations in the urban center, showed normal temperatures during the day and higher temperatures at night. Cluster 2, which had stations near the mountainous areas, showed much lower temperatures at night and slightly lower temperatures during the day. Cluster 3 was similar to Cluster 1, but the former exhibited slightly higher temperatures during the day and similar temperatures at night. The distribution of Cluster 3 was also similar to that of Cluster 1, but Cluster 3 was concentrated in the western part of Seoul, whereas Cluster 1 was evenly scattered throughout downtown Seoul. Further, Cluster 4 exhibited a slightly lower temperature at night, and was located between Clusters 1 and 2, and Cluster 5, which had stations in the center of the downtown areas, showed much higher temperatures during both the day and night. Cluster analyses implied that Seoul city could be classified into several climate zones according to the location of the station.

### 3.2. Coldwave Event Day (8 January 2021)

A coldwave event day was selected as 8 January 2021, which was the coldest day in Seoul in 2021. The average, maximum, and minimum temperatures in the Seoul ASOS station were recorded as −14.9, −10.7, and −18.6 °C, respectively.

Figure 6 shows the horizontal distribution of the daily mean, maximum, and minimum temperatures on the coldwave event day. The spatially averaged daily mean temperature was −12.9 °C, which was higher by 2.0 °C than that of the ASOS (Figure 6a). Further, 91% of the S-DoT stations showed a higher daily mean temperature than the temperature recorded at the ASOS, whereas only 9% showed lower mean values. The spatially averaged maximum temperature was −8.4 °C, which was higher by 2.3 °C than that recorded by the Seoul ASOS (Figure 6b). Moreover, 91% of the S-DoT stations exhibited a higher maximum temperature than the ASOS. The spatial average of the minimum temperature was −16.2 °C, which was higher by 1.6 °C than that recorded by the ASOS. Furthermore, 95% of the stations showed a higher minimum temperature than the ASOS, whereas only 5% showed a lower minimum temperature (Figure 6c).

The highest value among daily maximum temperatures was recorded as 5.8 °C at Changsin Station, the second hottest station during the heatwave event day (Figure 3a). A stovepipe exited heat energy 1 m apart on the same wall. The wall heated by direct solar radiation and heat energy from the stovepipe exit may have increased the sensor temperature abruptly. The sensor boxes at stations recorded as the top 3 maximum temperatures were all attached to walls. It can be concluded that the stations whose sensor boxes were attached to a wall of a building exhibited a much higher temperature than the other stations, implying that these could not represent the surrounding local climate zones. Thus, these stations should be removed before the basic quality control.

## 4. Quality Management System for S-DoT Meteorological Sensors (QMS-SDM)

QMS-SDM has pre-processing (Q0), basic quality control for a single station in real-time (Q1), extended quality control for multiple stations near real-time (Q2), and spatiotemporal gap-filling for multiple stations for daily data (Q3) steps (Figure 7). Each step had its own flag rule.

The stations installed on walls were removed before the QMS-SDM because the data did not represent the surrounding local climate zones, as mentioned in Section 3 (Figure 3). The first flag (A) addressed whether the observation environment was good or bad (QC00). If there were two first flags, the remaining flags became meaningless.

### 4.1. Pre-Processing (QC0)

Pre-processing has two steps: time allocation (QC01) and filling short missing data (QC02). The QC01 step allocates irregular sampling time to regular observation time every 2 min. First, the observation time was set to every 2 min from midnight; subsequently, the sampling time was allocated to the latest observation time in the backward direction.

Short-time data were missing owing to electrical problems or incomplete communication. Missing data shorter than 6 min (three data points) were filled using a linear regression equation with the most recent five data points. The autocorrelation function at 6 min is above 0.98 for air temperature and relative humidity.

### 4.2. Basic Quality Control for A Single Station in Real-Time (QC1)

The basic quality control for a single station in real-time has five quality checks: (1) physical limit check (QC11), (2) climate range check (QC12), (3) internal consistency check (QC13), (4) persistence check (QC14), and (5) step check (QC15). The physical limit check (QC11) excludes data with physically impossible values (WMO, 2002), whereas the climate range check (QC12) excludes climatological outlier values. Moreover, the upper (PL_u_ and CR_u_) and lower (PL_l_ and CR_l_) threshold values for physical limit and climate range checks should be determined in advance. The internal consistency check (QC13) detects and excludes data inconsistent with other data (WMO, 2002). The persistence check (QC14) detected data that do not vary with time. The most persistent data occurred because of electrical problems. The step check (QC15) detected data that varied abruptly. Moreover, the threshold values for QC14 and QC15 (PC and SC, respectively) were determined in advance.

Table 3 shows the lower and upper threshold values for physical limit checks applied to the ASOS/AWS operated by the KMA and AWS operated by local governments. The KMA and local governments used the same threshold values for relative humidity, wind speed, and direction. The AWS (local governments) applied −45 °C to the lower threshold values for air temperature, whereas the ASOS/AWS (KMA) applied −35 °C. The S-DoT sensors showed different lower and upper limits. The lower and upper limits were set to be −40 °C and 80 °C for air temperature, 0% and 100% for relative humidity, 0 m s^−1^ and 60 m s^−1^ for wind speed, 0° and 360° for wind direction, respectively (Table 3).

The lower and upper limits for climate range checks were determined according to the monthly climatology (Table 4) [19]. The KMA determined the lower and upper limits as the mean minus and plus *n* times the standard deviation, where *n* was 3–9 [9,10,19,20].

Most S-DoT sensors were installed downtown in Seoul. As a result, in Section 3, the upper temperature limits of the QMS-SDM were set higher than that of KMA by 7 °C (histograms in Figure 3 and Figure 6). The lower temperature limits of the QMS-SDM were set to be the same as those of the KMA.

The duration for the persistent test was set to 180 and 240 min for temperature and wind, respectively, both in the KMA and local governments (Table 5). As the duration already represents the longest time for the persistence check, the QMS-SDM used the same duration for the persistence test.

The allowable abrupt change for 1 min was set to 3 °C for temperature, 10% for relative humidity, and 10 m s^−1^ for wind speed in the KMA and local governments. Although the sampling rate of the QMS-SDM was 2 min, twice that of the ASOS and AWS, the QMS-SDM was set to have the same threshold values for step checks for temperature, relative humidity, and wind speed (Table 5).

### 4.3. Extended Quality Control for Multiple Stations near Real-Time (QC2)

The extended quality control for multiple stations near real-time comprises only one step: a spatial outlier check (QC21). The Madsen–Allerup method was applied to find spatial outlier data [21,22]. The median test statistic value $T_{it}$ is defined as:(1)$$T_{it}=\frac{x_{it}-M_{t}}{q_{t,75}-q_{t,25}}$$
where $x_{it}$ is the observation at station *i* at time *t*, $M_{t}\;$is the median value of the N observation at time *t*, and $q_{t,25}\;$and $q_{t,75}$ are the 25% and 75% quantile values of the N observations, respectively. If $|T_{it}|>2.0$, the data becomes unreliable. The air temperature was modified as a height-adjusted temperature considering the lapse rate of dry air temperature (9.8 °C km^−1^) before applying this step [23].

### 4.4. Data Reconstruction Using Spatiotemporal Gap-Filling for Daily Data (QC3)

QC3 comprised spatial gap filling (QC31) and temporal gap filling (QC32). The data removed during the QC22 step were reconstructed by averaging the values at the nearest three stations within 2 km. If only one or two stations were available, the values were replaced with the average values of one or two other stations. QC31 was applied to air temperature, relative humidity, and wind speed.

Temporal gap filling was applied to the data with missing periods of less than 30 min (15 data points). Further, the Stineman method was applied to fill in the missing data [24], and this method shows better performance for data with sudden slope changes and volatility clustering [25]. The method was selected based on the results of the performance test. The performance with respect to the imputation methods is provided in Appendix A.

### 4.5. Flag Rules

Each step has its own flag. In total, there were 10-digit flags. Character 0 represented the status “normal”, while Character 1 or 2 represented “not normal”. Flag 1 in Steps B, I, and J implied filled data, whereas Flag 1 in Steps E, F, G, and H implied doubtful or unreliable data. Flag 2 in Step A implied a bad observation environment, and Flag 2 in Steps C and D implied erroneous data (Table 6).

## 5. Application of the QMS-SDM to the S-DoT

The QMS-SDM was applied to 1-year data from August 2020 to July 2021. The 25 stations whose sensors were deployed on a building wall (Figure 3) were removed before QMS-SDM.

### 5.1. Distance between the Stations

The distances between the stations were calculated to consider the spatial gap-filling (QC3). Figure 8 shows a histogram of the shortest distance and the third-shortest distance from each station. The maximum frequency of the shortest distance occurred in the bin from 300 to 400 m. Moreover, 57.4% of the stations were at a distance of <400 m. The maximum frequency of the third-shortest distance occurred in the bin from 600 to 700 m.

### 5.2. Filling Short Missing Data

Most stations had many random missing data during a short period of less than 6 min. Figure 9a shows an example of the missing data ratio at every 2 h interval at a station on 27 July 2021. In total, there were 133 missing (NA, not available) temperature data points from 720 data points. Among them, there were 93 missing data for a single time and 17 missing data for 4 min (two consecutive times). Missing patterns were nearly random.

Entirely random data can be imputed to be good quality. Figure 9b shows that the imputed data fit well with the observed data. Daily data from a station were imputed using the Stineman method [24].

The daily missing data ratio depended on the day of the year and was extremely high (up to 80%) in April 2021, and low in the other months of 2021, except for some periods (Figure 10a). Missing ratios at most stations fell within the range of 10–20%. However, a sinusoidal periodic pattern was observed with respect to the station’s alphabetical order. This result may be due to data incomplete telecommunication failure from the site to the server. Thus, >98% of the missing data were imputed.

### 5.3. Basic Quality Control

Figure 11 shows an example of errors found using persistence and step checks. None of the data including temperature, relative humidity, PM_2.5_, and PM_10_ changed with time (Figure 11a). Contrastingly, all data were 0 at a given time (Figure 11b). Temporal differences in temperature (25 °C) and relative humidity (98%) were above the threshold values of 3 °C and 10%, respectively.

## 6. Summary and Conclusions

A quality management system for an IoT meteorological sensor network was developed. The QMS was applied to Seoul S-DoT, one of the largest and most diverse big data globally. The S-DoT sensors were installed over realistic urban areas, such as downtown areas, urban parks, and river sides, and not on ideal surfaces. The local climate zones near IoT network stations differed completely from those operated by the National Meteorological Administration [26,27]. Owing to irregular data transmission, the observation time was not regular. Some missing data were recorded as zero, which could not be distinguished from an observed zero value. Before the main quality control, a pre-processing step was added to unify the format.

The horizontal and temporal variations in temperature observed in the S-DoT network during heatwave and coldwave event days were investigated. Diurnal variation patterns observed in the S-DoT network were coherent with that obtained by the synoptic-scale ASOS station. The average, maximum, and minimum temperatures from the S-DoT were found to be higher than those from the ASOS on both event days. The temperatures at a few S-DoT stations were much higher than those at other stations because of the surrounding heating sources, such as walls. As these data could not represent surrounding local climate zones, they should be removed before the main QC.

Diurnal variation in temperature was classified into five clusters using DTW clustering. In general, stations with high temperatures during the daytime were located in the center of Seoul, whereas stations with low temperatures at night were located in forest areas near the city boundaries. Clustering analyses indicated that the upper and lower threshold values for climate range checks can be determined with respect to the cluster.

The QMS-SDM was designed to include two pre-processes, five basic quality controls, two extended quality controls, and two data reconstructions using gap filling. Quality control methodology and threshold values were defined based on previous studies and the present study. Normal, doubtful, and erroneous information for each QC step was saved as the corresponding flag file.

The QMS-SDM was applied to the S-DoT meteorological data from August 2020 to July 2021. Available data increased by 20% using the temporal imputation because the missing pattern was completely random. Using the QMS-SDM, data with irregular and diverse formats were changed to a regular and unified format. Furthermore, QC data are expected to provide high-quality and high-resolution urban meteorological information services effectively.

Moreover, the QMS-SDM can be applied to different IoT meteorological sensor networks. Pre-processing depends on the IoT sensor networks and should be reorganized. If the time interval is constant, then the synchronicity process may be unnecessary. Short missing data can be imputed using linear regression or the Stineman method. Furthermore, basic QC is essential for all networks. However, the threshold values should be redefined to match the sensor and climatology. Spatial outlier detection and spatial gap-filling may be useful for most networks.

## 7. Patents

The algorithm, QMS-SDM, is applied to the patent, entitled “Quality management system for IoT meteorological sensor network” (10-2002-0178038).

## Figures and Tables

**Figure 1 sensors-23-02384-f001:**
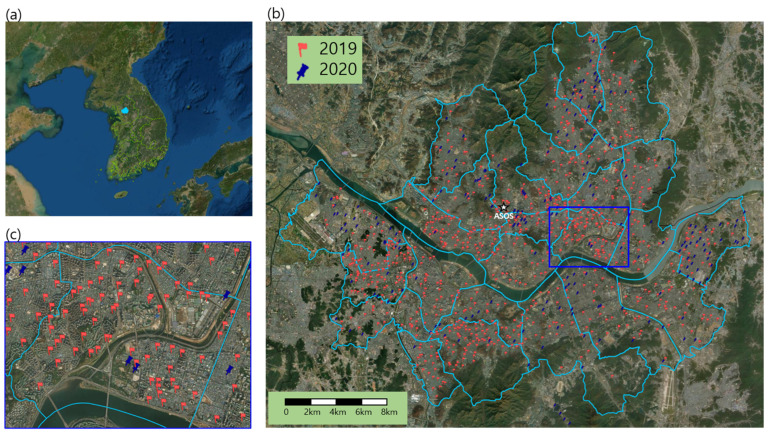
Location of (**a**) Seoul in Korea, (**b**) S-DoT stations installed in 2019 and 2020, and ASOS station (star) in Seoul City, and (**c**) enhanced location of S-DoT stations in Seongdong District (blue rectangle in (**b**)) in Seoul.

**Figure 2 sensors-23-02384-f002:**
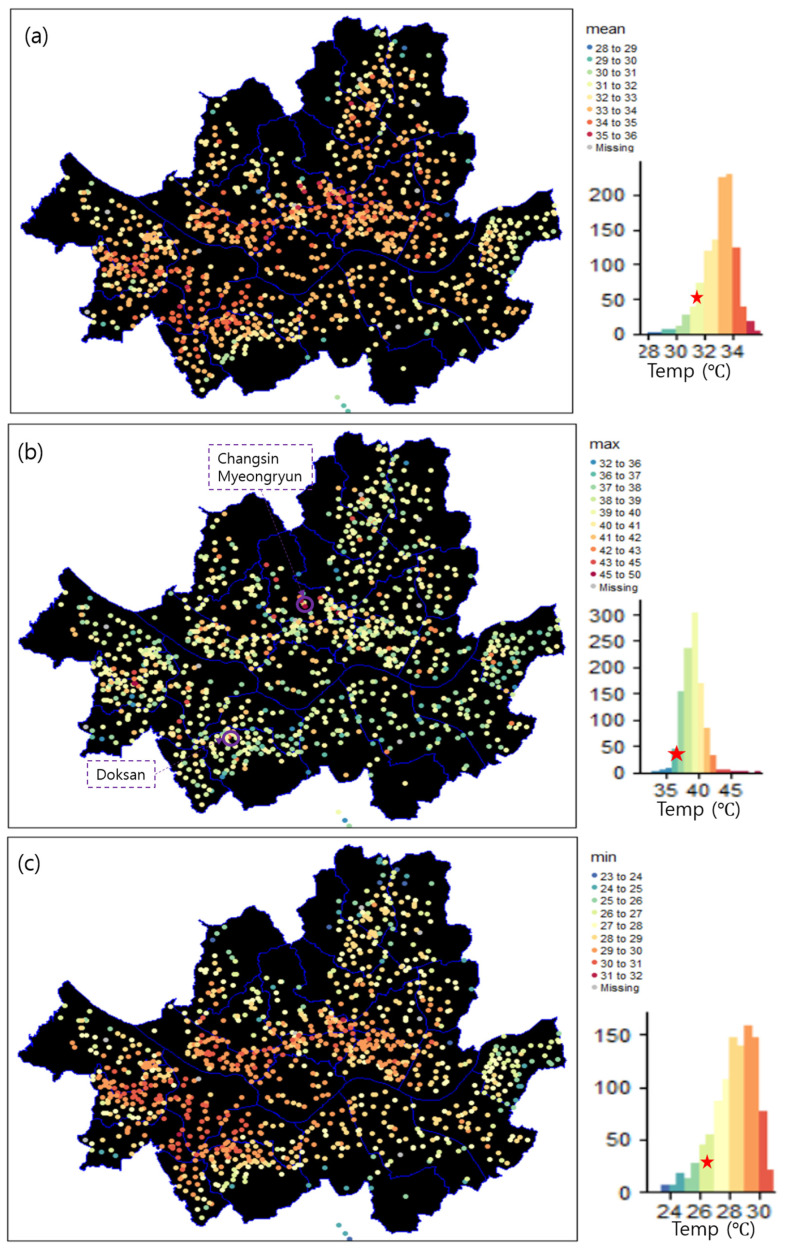
Horizontal distribution of daily (**a**) mean, (**b**) maximum, and (**c**) minimum temperatures on 24 July 2021, obtained by the S-DoT IoT meteorological sensors. The occurrence frequency for temperature is shown in the right bottom histogram. The values observed by Seoul ASOS are indicated by a red star.

**Figure 3 sensors-23-02384-f003:**
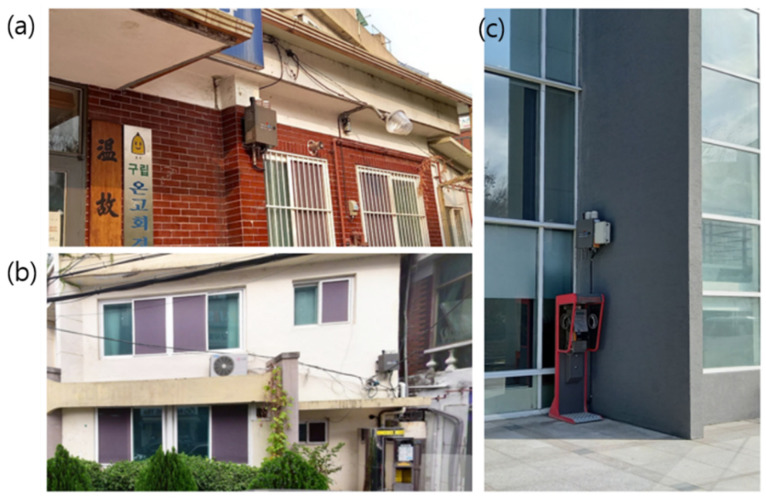
Sensor boxes in (**a**) Changsin-Dong, (**b**) Myeongryun-Dong, and (**c**) Doksan library stations that exhibited extreme temperatures.

**Figure 4 sensors-23-02384-f004:**
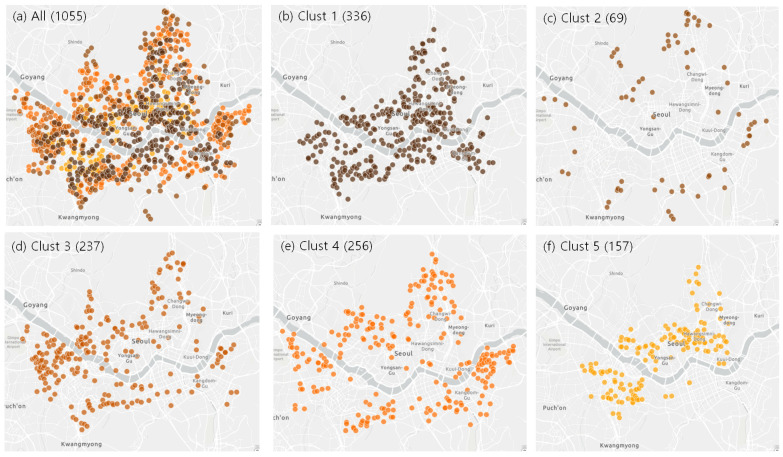
Location of (**a**) all stations and (**b**) Cluster 1, (**c**) Cluster 2, (**d**) Cluster 3, (**e**) Cluster 4, and (**f**) Cluster 5 stations.

**Figure 5 sensors-23-02384-f005:**
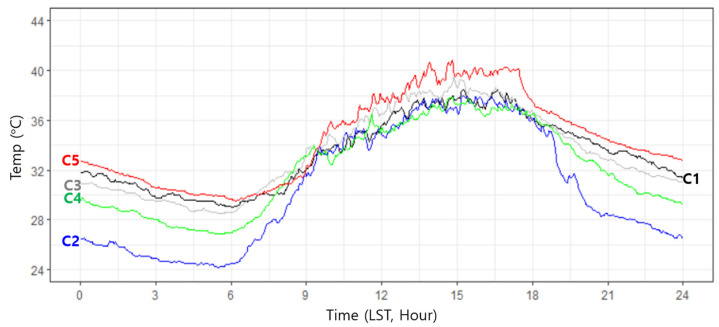
Time series of centroid temperature for each cluster.

**Figure 6 sensors-23-02384-f006:**
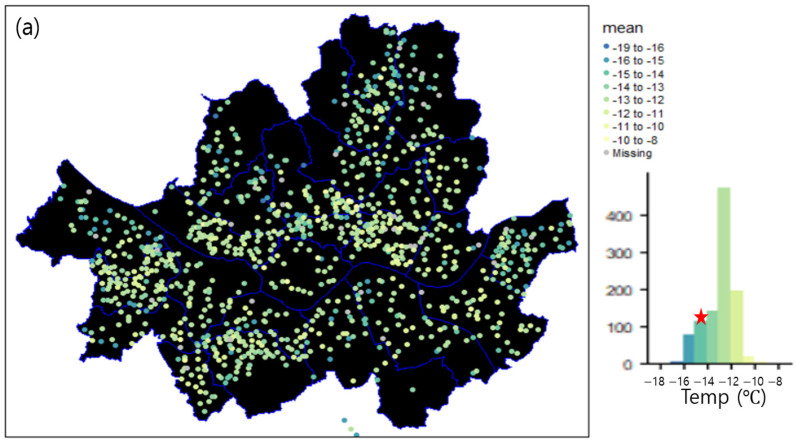
Horizontal distribution of daily (**a**) mean, (**b**) maximum, and (**c**) minimum temperatures on 8 January 2021, obtained by the S-DoT IoT meteorological sensors. The occurrence frequency for temperature is shown in the right bottom histogram. The values observed by Seoul ASOS are indicated by a red star.

**Figure 7 sensors-23-02384-f007:**
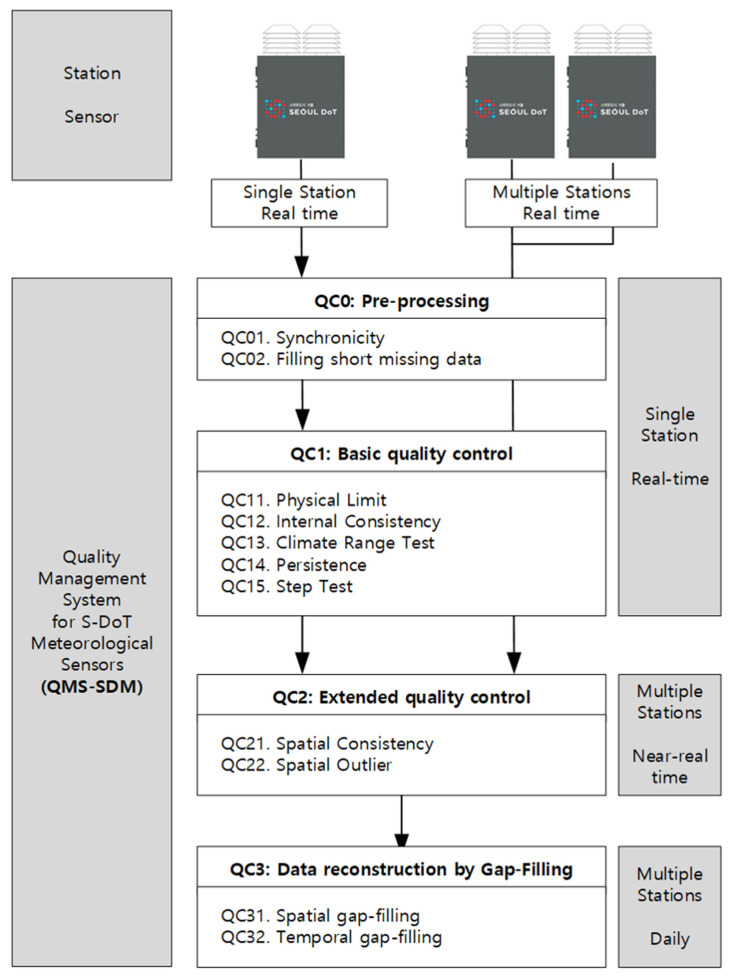
Flow chart of the quality management system for the S-DoT meteorological sensors (QMS-SDM).

**Figure 8 sensors-23-02384-f008:**
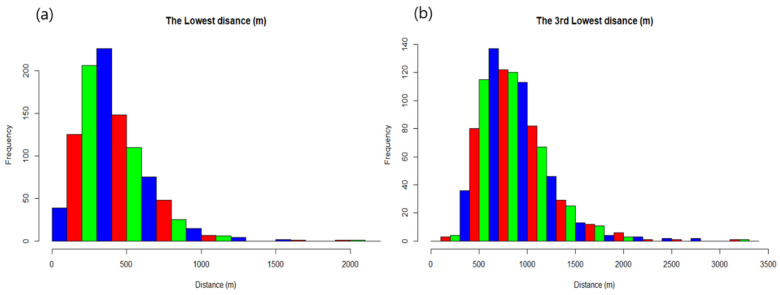
Histogram of (**a**) the shortest and (**b**) the third-shortest distance between two stations.

**Figure 9 sensors-23-02384-f009:**
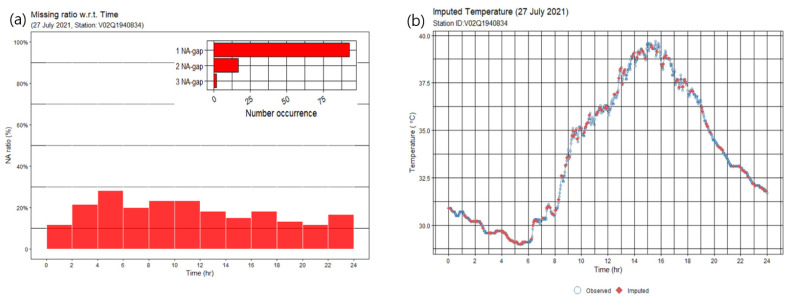
(**a**) Histogram of the number of NA (Not Available) data with respect to the time of a day, and (**b**) time series of observed (blue open circle) and imputed (red diamond) data on 27 July 2021 at a station.

**Figure 10 sensors-23-02384-f010:**
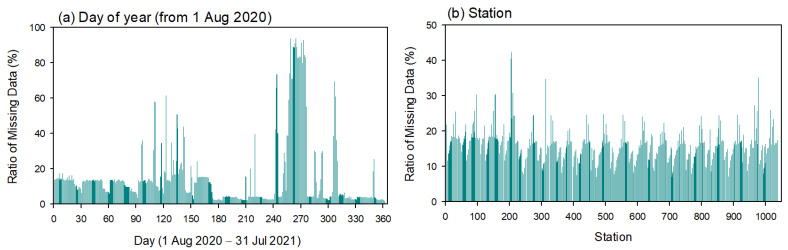
Ratio of missing data with respect to (**a**) day of the year and (**b**) station alphabetical order.

**Figure 11 sensors-23-02384-f011:**
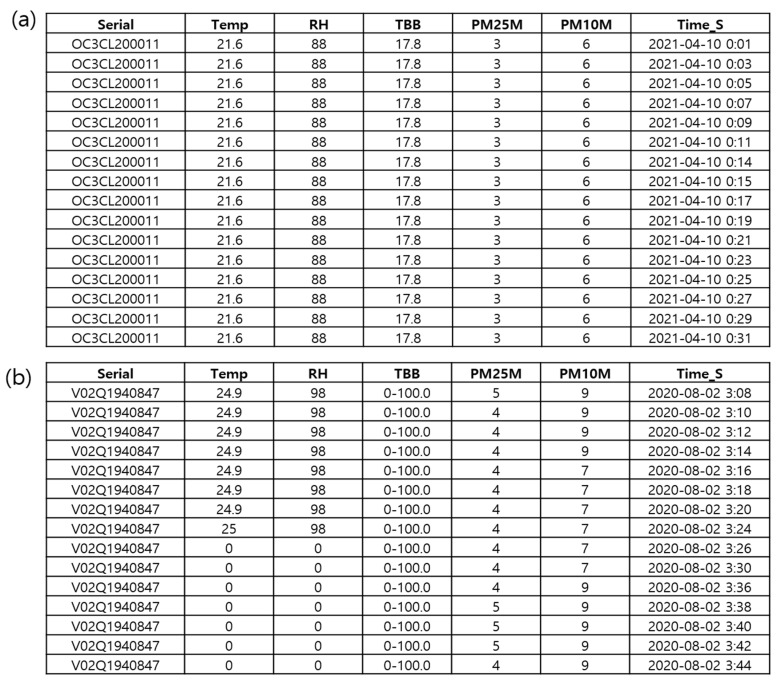
Example of erroneous data found using the (**a**) persistence and (**b**) step check.

**Table 1 sensors-23-02384-t001:** Number of stations with respect to the surrounding environments.

Surrounding Environment	Number of Stations	Remarks
Downtown	923 (25 *)	Streetlight, CCTV, smart poll, walls
Riverside	42	
Mountain	99	

* Number of stations installed on walls. CCTV: closed circuit television.

**Table 2 sensors-23-02384-t002:** Installed sensors with respect to the S-DoT station types.

Station Location	Sensors	Number of Stations
Basic (Type A)	Temperature, relative humidity, noise, illumination, ultraviolet, vibration, PM_10_, PM_2.5_	992
Type B	Basic + wind speed, direction or globe temperature	32
Type C	Basic + CO, NO_2_, SO_2_	20
Type D	Basic + NH_3_, H_2_S, O_3_	20

**Table 3 sensors-23-02384-t003:** Lower and upper threshold values for physical limit checks applied to the ASOS/AWS operated by KMA, AWS operated by the local government, and QMS-SDM in Korea.

Sensors	ASOS/AWS (KMA)		AWS (Local Gov.)		QMS-SDM	
Temperature (°C)	−35	45	−45	45	−40	80
Relative humidity (%)	1	100	1	100	0	100
Wind Speed (m s^−1^)	0	75	0	75	0	60
Wind direction (°)	1	360	1	360	0	360

**Table 4 sensors-23-02384-t004:** Threshold values (°C) for air temperature climate range checks applied to the KMA ASOS/AWS, AWS by the local government, and QMS-SDM in Korea.

Month	ASOS/AWS (KMA)		AWS (Local Gov.)		QMS-SDM	
January	−33	25	−35	30	−33	32
February	−31	30	−35	30	−31	37
March	−26	35	−30	35	−26	42
April	−21	41	−20	40	−21	48
May	−12	44	−10	45	−12	51
June	2	44	0	45	2	51
July	5	45	5	45	5	52
August	4	45	0	45	4	52
September	−6	43	−10	45	−6	50
October	−17	42	−20	40	−17	49
November	−26	36	−35	35	−26	43
December	−29	27	−35	30	−29	34

**Table 5 sensors-23-02384-t005:** Threshold values for persistence and step checks applied to the ASOS/AWS operated by KMA, AWS operated by the local government, and QMS-SDM in Korea.

Sensors	Persistence (min)			Step (/(1 or 2 min))		
	ASOS/ AWS (KMA)	AWS (Local)	QMS-SDM	ASOS/ AWS (KMA)	AWS (Local)	QMS-SDM
Temperature	180	180	180	3 °C	3 °C	3 °C
Relative humidity	-	-	-	-	10%	10%
Wind Speed	-	240	240	10 m s^−1^	10 m s^−1^	10 m s^−1^
Wind direction	240	240	240	-	-	-

**Table 6 sensors-23-02384-t006:** Flag rules for QMS-SDM.

A	B	C	D	E	F	G	H	I	J
A (QC00)			0: Normal			2: Bad observation environment			
B (QC02)			0: Normal			1: Filled			
C (QC11)			0: Normal			2: Erroneous			
D (QC12)			0: Normal			2: Erroneous			
E (QC13)			0: Normal			1: Doubtful or unreliable			
F (QC14)			0: Normal			1: Doubtful or unreliable			
G (QC15)			0: Normal			1: Doubtful or unreliable			
H (QC21)			0: Normal			1: Doubtful or unreliable			
I (QC31)			0: Observed			1: Gap-filled			
J (QC32)			0: Observed			1: Gap-filled			

## Data Availability

The QMS-SDM algorithm can be provided upon request. For further inquiries, please contact Moon-Soo Park (moonsoo@sejong.ac.kr).

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
