# Peer review of "Quality Management System for an IoT Meteorological Sensor Network—Application to Smart Seoul Data of Things (S-DoT)"

_sensors, 2023, doi:10.3390/s23052384_

Round 1

Reviewer 1 Report

Report 1

The present manuscript seeks to apply quality control to sensors included in an urban network that provides data at a suitable resolution for phenomena at the scale of the city, and particularly at the scale of megacities as Seoul. No doubt, the manuscript contains some useful information, it has novel insights due to the use of an advanced data transmission system, and it wants to verify the statistical consistency and robustness of the data provided by the new network against the criteria recommended by WMO for sensors and in this respect, it would be worthwhile to consider for a publication.

However, to aggregate and use S-DoT measurements considering its space-time variability because of heat waves , which result from synoptic-scale climate change with reflections at urban scales, one must consider a statistic of extreme events over which to extend climate limits. A single extreme heat/cold event does not constitute a sufficient statistic also considering that the clustering technique used is applied without difference in the vertical domain to all stations and the resulting statistics made only of histograms  is not readable .

Then considering that national and local weather service data will certainly have recorded such events over the past decade the comparison of more accurate statistics of extreme events (heat waves and cold spells) with the case chosen as a test example could help to understand whether the S-DoT network can measure climate variability within the city by extending threshold values used as climate limits.

With these considerations, I suggest returning the present manuscript with a major revision.

The article is rambling and inaccurate in some places, which need to be corrected.

More Specific Presentation Issues:

As a first observation, not having been able to read reference [2], in Korean (line35), and the supplementary material (missing the supplementary file) does not help the understanding of the article. In addition, references (line 31-51) should be expanded for example including urban observations since the topic is related to observations quality control in urban environment, contained in an Annex to Chapter 9 of the Guide to the WMO Integrated Global Observing System (WMO, 2018).

(Line 33) What is the network purpose? The climate limits that one would apply to detect phenomena such as heat waves and their effect on urban climate are related to the scale of heat waves and their variability. If the network to be tested is used for air quality detection, the limits are related to different meteorological phenomena, and the resulting placement of stations (sensors) is different. Authors should specify and use climate limits congruent with the purpose of the network.

(Line 138-184)The urban environment as it is defined in the WMO Guide, is such that most developed sites make it impossible to conform to the standard guidelines for site selection and instrument exposure. The idea of measuring various weather phenomena as the paper starts, is connected to measuring weather parameters in the urban environment where the horizontal scales extend from meters to tents of kilometers in the horizontal direction, but with the fundamental distinction between the height of BL, UCL and the sublayer where the effects of turbulent eddies mix microclimatic effects of the buildings, obstacles and fluxes of moisture and heat are relevant.

Then in order to aggregate and use the proposed network, before QC, one must make sure that the S-DoT is in the right environments and specify if this is not the case and distinguish between them; clustering does not explain this and does not categorize the stations according to the scales that characterize them.

Figure 2: beside the meaning of this statistics the histogram is not readable, values are blurred and not understandable. The colors of the different temperatures and the caption say that we are looking to daily mean maximum and minimum of temperatures, so the frequency of the network that this histogram show is the frequency of occurrence of these temperatures for the two (2) case studied? The red star is supposed to show that the values are extreme with respect the ASOS values? Which values, climatological, mean values of extreme events or? Explain and show.

Figure 6 Same as Fig 2 not readable. Moreover, to deleted stations from a statistic that is tested , you must apply the QC, it is precisely by using quality control applied in the most correct manner that extreme temperature values resulting from improper station placement  can be detected and highlight network defects, which in this way can be corrected with a new positioning of sensors (Line 217-222). Then you can flag, and eventually erase, data.

(Line 281) histograms are in Fig 2 and 6, not 3 and 6. The 7 degrees rise of KMA limit is correct as far as the network is measuring a correct bias related to heat wave statistic.

Line 307 Correction of temperature vertically with the adiabatic lapse rate is very questionable in BL in general , due to the presence of moisture and heat fluxes, that is enthalpy, which are even less adiabatic in an urban environment. But this correction is related to the height of the sensors that have already been mixed in the previous statistic.

Line 319 Where are Table S1 and S2? Not able to recover them.

Line 356-362 Clarify: what is this pattern of errors and why it is irrelevant.

Line 366-372 Fig. 11 Not readable, not understandable.

Line 391-395 Diurnal variations have been considered and a pattern emerged, is this coherent with the heat wave statistics on larger scales?

Line 408-413 The QC elaborated can be applied to other meteorological sensor network whether the criteria used in the QC meet the WMO criteria for measuring temperature appropriately by correlating the purpose of the network with the phenomena to be measured. This generalization cannot be made by the method used.

Author Response

Replies to Report 1

Thank you for reviewers’ thorough and valuable comments. Authors replied to all comments in Blue. Major corrections were indicated in red in the revised manuscript. Due to reviewer’s comments, the manuscript was much improved.

The present manuscript seeks to apply quality control to sensors included in an urban network that provides data at a suitable resolution for phenomena at the scale of the city, and particularly at the scale of megacities as Seoul. No doubt, the manuscript contains some useful information, it has novel insights due to the use of an advanced data transmission system, and it wants to verify the statistical consistency and robustness of the data provided by the new network against the criteria recommended by WMO for sensors and in this respect, it would be worthwhile to consider for a publication.

  • Thank you for your comments.

However, to aggregate and use S-DoT measurements considering its space-time variability because of heat waves , which result from synoptic-scale climate change with reflections at urban scales, one must consider a statistic of extreme events over which to extend climate limits. A single extreme heat/cold event does not constitute a sufficient statistic also considering that the clustering technique used is applied without difference in the vertical domain to all stations and the resulting statistics made only of histograms is not readable .

  • Heatwave and coldwave events are one of synoptic-scale weather phenomena. But urban areas tend to show higher temperature during the heatwave period due to extra heat storage in and release from materials such as concrete and asphalt. The above description was added in L147-151 in the revised manuscript.
  • Histogram were redrawn to be large and more readable in Figs. 2 and 6.

Then considering that national and local weather service data will certainly have recorded such events over the past decade the comparison of more accurate statistics of extreme events (heat waves and cold spells) with the case chosen as a test example could help to understand whether the S-DoT network can measure climate variability within the city by extending threshold values used as climate limits.

With these considerations, I suggest returning the present manuscript with a major revision.

The article is rambling and inaccurate in some places, which need to be corrected.

  • Reliable meteorological measurements in Seoul City were confined to the 1 ASOS and 25 AWS stations operated by KMA. But above half of these were installed over a rooftop on a building, the remaining ones were installed over suburban surfaces and not realistic urban surfaces. On the other hands, most S-DoT stations were installed over realistic urban areas surrounded by middle- or high-rise compact buildings. The above expression was added in L103-107 in the revised manuscript.
  • This manuscript did not concentrate on statistics during the heatwave and coldwave events, but the possibility of higher temperature at the realistic urban areas than at the pseudo-urban areas.

More Specific Presentation Issues:

As a first observation, not having been able to read reference [2], in Korean (line35), and the supplementary material (missing the supplementary file) does not help the understanding of the article. In addition, references (line 31-51) should be expanded for example including urban observations since the topic is related to observations quality control in urban environment, contained in an Annex to Chapter 9 of the Guide to the WMO Integrated Global Observing System (WMO, 2018).

  • Abstract, as well as all Figures and Tables in Reference [2] were expressed in English. These might be helpful to understand the reference.
  • WMO (2018) was referred as WMO (2021b), updated version of the same document. Moreover, Supplement Tables are attached to the end of main text.

(Line 33) What is the network purpose? The climate limits that one would apply to detect phenomena such as heat waves and their effect on urban climate are related to the scale of heat waves and their variability. If the network to be tested is used for air quality detection, the limits are related to different meteorological phenomena, and the resulting placement of stations (sensors) is different. Authors should specify and use climate limits congruent with the purpose of the network.

  • The S-DoT aimed not only to assist municipal policies related to living quality such as air quality, noise, vibration, odors, but also to deliver various useful information including meteorological variables to their citizen. The above expression was added in L98-100 in the revised manuscript.

(Line 138-184)The urban environment as it is defined in the WMO Guide, is such that most developed sites make it impossible to conform to the standard guidelines for site selection and instrument exposure. The idea of measuring various weather phenomena as the paper starts, is connected to measuring weather parameters in the urban environment where the horizontal scales extend from meters to tents of kilometers in the horizontal direction, but with the fundamental distinction between the height of BL, UCL and the sublayer where the effects of turbulent eddies mix microclimatic effects of the buildings, obstacles and fluxes of moisture and heat are relevant.

  • We should take into account horizontal distribution as well as vertical cross sections. But this manuscript focus on the horizontal distribution of surface meteorological variables, especially of surface temperature, not vertical structures such as BL, UCL.
  • Heatwave event has become more frequent and been strong in Seoul City due to climate change. The patients with heat patient have been increased.

Then in order to aggregate and use the proposed network, before QC, one must make sure that the S-DoT is in the right environments and specify if this is not the case and distinguish between them; clustering does not explain this and does not categorize the stations according to the scales that characterize them.

  • We’ve checked installation environment for all stations. Through the installation environment check, some stations installed on walls were removed before the QMS-SDM because the data could not represent the surrounding local climate zones. The above were described in L232-236 and 243-246 in the revised manuscript.

Figure 2: beside the meaning of this statistics the histogram is not readable, values are blurred and not understandable. The colors of the different temperatures and the caption say that we are looking to daily mean maximum and minimum of temperatures, so the frequency of the network that this histogram show is the frequency of occurrence of these temperatures for the two (2) case studied? The red star is supposed to show that the values are extreme with respect the ASOS values? Which values, climatological, mean values of extreme events or? Explain and show.

  • Histogram was redrawn to be large. Explanation on the figures was found in the caption: Horizontal distribution of daily (a) mean, (b) maximum, and (c) minimum temperatures on 24 July 2021, obtained by the S-DoT IoT meteorological sensors. The occurrence frequency for temperature is shown in the right bottom histogram. The values observed by Seoul ASOS are indicated by a red star.

Figure 6 Same as Fig 2 not readable. Moreover, to deleted stations from a statistic that is tested , you must apply the QC, it is precisely by using quality control applied in the most correct manner that extreme temperature values resulting from improper station placement can be detected and highlight network defects, which in this way can be corrected with a new positioning of sensors (Line 217-222). Then you can flag, and eventually erase, data.

  • Through the horizontal distribution in Section 3, the stations whose sensor boxes were attached to wall of a building were removed, as mentioned in L243-246 in the revised manuscript.

(Line 281) histograms are in Fig 2 and 6, not 3 and 6. The 7 degrees rise of KMA limit is correct as far as the network is measuring a correct bias related to heat wave statistic.

  • 3 is changed to Fig. 2.

Line 307 Correction of temperature vertically with the adiabatic lapse rate is very questionable in BL in general , due to the presence of moisture and heat fluxes, that is enthalpy, which are even less adiabatic in an urban environment. But this correction is related to the height of the sensors that have already been mixed in the previous statistic.

  • Generally, the environmental lapse rate is about 6°C/km, which is lower than the dry adiabatic lapse rate of 9.8°C/km. But the dry adiabatic lapse rate is adapted from Sheridan et al.(2010). The above was described in L319-321.

Line 319 Where are Table S1 and S2? Not able to recover them.

  • Supplementary materials were attached to the P18 in the revised manuscript.

Line 356-362 Clarify: what is this pattern of errors and why it is irrelevant.

  • The expression was modified to clarify the meaning such that: Missing ratios at most stations fell within the range of 10-20%, However, a sinusoidal periodic pattern was observed with respect to the station’s alphabetical order. This result may be due to data incomplete telecommunication failure from the site to server. The description was in L372-375.

Line 366-372 Fig. 11 Not readable, not understandable.

  • 11 was redrawn in the revised manuscript.

Line 391-395 Diurnal variations have been considered and a pattern emerged, is this coherent with the heat wave statistics on larger scales?

  • Diurnal variation pattern of S-DoT stations were coherent with that obtained by the synoptic-scale ASOS station. The above expression was added in L400-402.

Line 408-413 The QC elaborated can be applied to other meteorological sensor network whether the criteria used in the QC meet the WMO criteria for measuring temperature appropriately by correlating the purpose of the network with the phenomena to be measured. This generalization cannot be made by the method used.

  • After pre-processing, QMS-SDM can be applied to other high-resolution meteorological network. The above description was in L425-428.

Reviewer 2 Report

Dear Authors

I read the paper: Quality Management System for an IoT Meteorological Sensor Network – Application to Smart Seoul Data of Things (S-DoT). After reading I have some suggestions and advice. I hope it will be useful for the Authors.

I don’t understand some parts of the sentence - What does “a 10-m horizontal scale” mean? and “1-min temporal scale”?

What Authors wanted to say by this sentence: An array of things developed in Chicago is an example?

Line 33 – Authors mentioned about some weather phenomena, but there is no drought. Is it correct that drought were not mentioned? In the papers: Nexus between water, energy, food and climate change as challenges facing the modern global, European and Polish economy or/and Climate change and extreme weather events: can developing countries adapt? Authors can read about many aspects of weather condition, climate changes and their influence on economy. There is nexus between mentioned elements. Similar in Korea IoT or S-DoT will help to control (maybe predict) the weather changes?

Line 60 – why Authors in one paragraph mentioned about Nordic countries (e.g. Finland, Sweden, Norway) and Oklahoma in USA? Please write something, which connect those examples.

Line 159 and 213 – Is it correct? “The maximum daily maximum temperature”

Table 3 and Table 5 – please make the same style of the tables. In one table units are in the text in other table in the first column.

Please make better quality of Figure 11.

Please make conclusion section.

Author Response

Replies to Report 2

Thank you for reviewers’ thorough and valuable comments. Authors replied to all comments in Blue. Major corrections were indicated in red in the revised manuscript. Due to reviewer’s comments, the manuscript was much improved.

Report 2

I read the paper: Quality Management System for an IoT Meteorological Sensor Network – Application to Smart Seoul Data of Things (S-DoT). After reading I have some suggestions and advice. I hope it will be useful for the Authors.

I don’t understand some parts of the sentence - What does “a 10-m horizontal scale” mean? and “1-min temporal scale”?

  • Most building in cities has a 10-m order horizontal scale. Street canyon has a scale of 10-m horizontal scale. Surface temperature was different with respect to each facet, sunny side, or shadow side. Surface temperature in cities can change by large values in 1 minute.

What Authors wanted to say by this sentence: An array of things developed in Chicago is an example?

  • “Array of Things” was a name of sensor network constructed in Chicago. The expression was modified to clarify the meaning such that: The “Array of Things” project developed in Chicago is an example [L39-40]

Line 33 – Authors mentioned about some weather phenomena, but there is no drought. Is it correct that drought were not mentioned? In the papers: Nexus between water, energy, food and climate change as challenges facing the modern global, European and Polish economy or/and Climate change and extreme weather events: can developing countries adapt? Authors can read about many aspects of weather condition, climate changes and their influence on economy. There is nexus between mentioned elements. Similar in Korea IoT or S-DoT will help to control (maybe predict) the weather changes?

  • Drought was added in weather phenomena in L33 in the revised manuscript. A reference entitled with “Nexus between water, energy, food and climate change….” was added in L33-34 and [5].

Line 60 – why Authors in one paragraph mentioned about Nordic countries (e.g. Finland, Sweden, Norway) and Oklahoma in USA? Please write something, which connect those examples.

  • Most quality controls for meteorological data were governed by WMO guidance. Nordic and USA were examples of well documented QC algorithm.

Line 159 and 213 – Is it correct? “The maximum daily maximum temperature”

  • It is correct. But the expression was changed to “heighest value among daily maximum temperature” in L174 and 227 in the revised manuscript.

Table 3 and Table 5 – please make the same style of the tables. In one table units are in the text in other table in the first column.

  • Left and right panels in Table 5 has different units of minute and value dependent ones such as °C, %, and m/s.

Please make better quality of Figure 11.

  • Figure 11 was improved.

Please make conclusion section.

  • Chapter 6 has summary, discussion, and conclusion. So the title “Summary and Discussions” was changed to “Summary and Conclusions”.

Round 2

Reviewer 1 Report

This paper is about a high-resolution sensor network providing information that can be useful to citizens for different purposes. Usefulness as a network for climatic and other phenomena requires more complex processing, but since the network has not been captured in any global network, it can be accepted as long as it is clear in the introduction and conclusions that the S-DoT sensors are valid if threshold values after QC are redefined to match climatology, even in the wave cases examined.

Replies to Report 1

Thank you for reviewers’ thorough and valuable comments. Authors replied to all comments in Blue. Major corrections were indicated in red in the revised manuscript. Due to reviewer’s comments, the manuscript was much improved.

The present manuscript seeks to apply quality control to sensors included in an urban network that provides data at a suitable resolution for phenomena at the scale of the city, and particularly at the scale of megacities as Seoul. No doubt, the manuscript contains some useful information, it has novel insights due to the use of an advanced data transmission system, and it wants to verify the statistical consistency and robustness of the data provided by the new network against the criteria recommended by WMO for sensors and in this respect, it would be worthwhile to consider for a publication.

ð  Thank you for your comments.

However, to aggregate and use S-DoT measurements considering its space-time variability because of heat waves , which result from synoptic-scale climate change with reflections at urban scales, one must consider a statistic of extreme events over which to extend climate limits. A single extreme heat/cold event does not constitute a sufficient statistic also considering that the clustering technique used is applied without difference in the vertical domain to all stations and the resulting statistics made only of histograms is not readable .

ð  Heatwave and coldwave events are one of synoptic-scale weather phenomena. But urban areas tend to show higher temperature during the heatwave period due to extra heat storage in and release from materials such as concrete and asphalt. The above description was added in L147- 151 in the revised manuscript.

ð  Histogram were redrawn to be large and more readable in Figs. 2 and 6.

The reply does not change the fact that the extra heat storage is superimposed to an urban effect due to the urban features like heat fluxes due to materials, as you have written. The base of the heat storage is due to the urban climate that is not quantified in this paper, despite the sensors added, and then the limits used for the QC are arbitrary.

Then considering that national and local weather service data will certainly have recorded such events over the past decade the comparison of more accurate statistics of extreme events (heat waves and cold spells) with the case chosen as a test example could help to understand whether the S-DoT network can measure climate variability within the city by extending threshold values used as climate limits.

With these considerations, I suggest returning the present manuscript with a major revision. The article is rambling and inaccurate in some places, which need to be corrected.

ð  Reliable meteorological measurements in Seoul City were confined to the 1 ASOS and 25 AWS stations operated by KMA. But above half of these were installed over a rooftop on a building, the remaining ones were installed over suburban surfaces and not realistic urban surfaces. On the other hands, most S-DoT stations were installed over realistic urban areassurrounded by middle- or high-rise compact buildings. The above expression was added in L103-107 in the revised manuscript.

ð  This manuscript did not concentrate on statistics during the heatwave and coldwave events, but the possibility of higher temperature at the realistic urban areas than at the pseudo-urban areas.

The work under discussion does not consider the statistic it is using to do quality control but uses sensor classification as a criterion to change the climate limits used arbitrarily based on a single event being measured, further extending the usefulness of this statistic to extreme events of very different types such as intense and sudden rainfall, even less related to the heat wave and cold wave event used as examples.

More Specific Presentation Issues:

As a first observation, not having been able to read reference [2], in Korean (line35), and the supplementary material (missing the supplementary file) does not help the understanding of the article. In addition, references (line 31-51) should be expanded for example including urban observations since the topic is related to observations quality control in urban environment, contained in an Annex to Chapter 9 of the Guide to the WMO Integrated Global Observing System (WMO, 2018).

ð  Abstract, as well as all Figures and Tables in Reference [2] were expressed in English. These might be helpful to understand the reference.

Reference 2 was not in English and supplementary materials were missing, but this is a minor failure of the paper, although it is very disrespectful of the reviewers not to allow them to do the work of reviewing.

ð  WMO (2018) was referred as WMO (2021b), updated version of the same document. Moreover, Supplement Tables are attached to the end of main text.

(Line 33) What is the network purpose? The climate limits that one would apply to detect phenomena such as heat waves and their effect on urban climate are related to the scale of heat waves and their variability. If the network to be tested is used for air quality detection, the limits are related to different meteorological phenomena, and the resulting placement of stations (sensors) is different. Authors should specify and use climate limits congruent with the purpose of the network.

ð  The S-DoT aimed not only to assist municipal policies related to living quality such as air quality, noise, vibration, odors, but also to deliver various useful information including meteorological variables to their citizen. The above expression was added in L98-100 in the revised manuscript.

This is the only real reply of the authors: your paper is about a local network of sensors measuring different quantities on a local space and time scale.

(Line 138-184)The urban environment as it is defined in the WMO Guide, is such that most developed sites make it impossible to conform to the standard guidelines for site selection and instrument exposure. The idea of measuring various weather phenomena as the paper starts, is connected to measuring weather parameters in the urban environment where the horizontal scales extend from meters to tents of kilometers in the horizontal direction, but with the fundamental distinction between the height of BL, UCL and the sublayer where the effects of turbulent eddies mix microclimatic effects of the buildings, obstacles and fluxes of moisture and heat are relevant.

ð  We should take into account horizontal distribution as well as vertical cross sections. But this manuscript focus on the horizontal distribution of surface meteorological variables, especially of surface temperature, not vertical structures such as BL, UCL.

The horizontal and vertical scale in the urban environment as well as in general are connected trough the fluxes of enthalpy (temperature and humidity), momentum (velocities) that in  this case are turbulent. Then the temperature horizontal scale is connected to the vertical distribution of all these quantities, when you change the climate limits you change the space and time scale you are monitoring.

ð  Heatwave event has become more frequent and been strong in Seoul City due to climate change. The patients with heat patient have been increased.

Then in order to aggregate and use the proposed network, before QC, one must make sure that the S- DoT is in the right environments and specify if this is not the case and distinguish between them; clustering does not explain this and does not categorize the stations according to the scales that characterize them.

ð  We’ve checked installation environment for all stations. Through the installation environment check, some stations installed on walls were removed before the QMS-SDM because the data could not represent the surrounding local climate zones. The above were described in L232- 236 and 243-246 in the revised manuscript.

These are stations checks, first steps to build any network. You are mentioning it, but to cluster them in the horizontal following a single case (the hot or cold wave) does not make the environment homogeneous. Any way this step is the only acceptable from a general QC

Figure 2: beside the meaning of this statistics the histogram is not readable, values are blurred and not understandable. The colors of the different temperatures and the caption say that we are looking to daily mean maximum and minimum of temperatures, so the frequency of the network that this histogram show is the frequency of occurrence of these temperatures for the two (2) case studied? The red star is supposed to show that the values are extreme with respect the ASOS values? Which values, climatological, mean values of extreme events or? Explain and show.

ð  Histogram was redrawn to be large. Explanation on the figures was found in the caption: Horizontal distribution of daily (a) mean, (b) maximum, and (c) minimum temperatures on 24 July 2021, obtained by the S-DoT IoT meteorological sensors. The occurrence frequency for temperature is shown in the right bottom histogram. The values observed by Seoul ASOS are indicated by a red star.

Figure 6 Same as Fig 2 not readable. Moreover, to deleted stations from a statistic that is tested , you must apply the QC, it is precisely by using quality control applied in the most correct manner that extreme temperature values resulting from improper station placement can be detected and highlight network defects, which in this way can be corrected with a new positioning of sensors (Line 217-222). Then you can flag, and eventually erase, data.

ð  Through the horizontal distribution in Section 3, the stations whose sensor boxes were attached to wall of a building were removed, as mentioned in L243-246 in the revised manuscript.

(Line 281) histograms are in Fig 2 and 6, not 3 and 6. The 7 degrees rise of KMA limit is correct as far as the network is measuring a correct bias related to heat wave statistic.

ð  Fig. 3 is changed to Fig. 2Line 307 Correction of temperature vertically with the adiabatic lapse rate is very questionable in BL in general , due to the presence of moisture and heat fluxes, that is enthalpy, which are even less adiabatic in an urban environment. But this correction is related to the height of the sensors that have already been mixed in the previous statistic.

ð  Generally, the environmental lapse rate is about 6°C/km, which is lower than the dry adiabatic lapse rate of 9.8°C/km. But the dry adiabatic lapse rate is adapted from Sheridan et al.(2010). The above was described in L319-321.

Sheridan et al.(2010) is discussing a correction of lapse rate due to different orographic position in a large region where the difference is due to mountain. The potential temperature difference in this case is due to different pression at different heights. Not apt to this application.

The differences in the urban environment are better measured with virtual potential temperature measuring the humidity and temperature differences due to different fluxes because of materials but also to buildings and currents (turbulent fluxes) between them and other features of this kind.

Line 319 Where are Table S1 and S2? Not able to recover them.

ð  Supplementary materials were attached to the P18 in the revised manuscript.

Line 356-362 Clarify: what is this pattern of errors and why it is irrelevant.

ð  The expression was modified to clarify the meaning such that: Missing ratios at most stations fell within the range of 10-20%, However, a sinusoidal periodic pattern was observed with respect to the station’s alphabetical order. This result may be due to data incomplete telecommunication failure from the site to server. The description was in L372-375.

Line 366-372 Fig. 11 Not readable, not understandable.

ð  Fig. 11 was redrawn in the revised manuscript.

Line 391-395 Diurnal variations have been considered and a pattern emerged, is this coherent with the heat wave statistics on larger scales?

ð  Diurnal variation pattern of S-DoT stations were coherent with that obtained by the synoptic- scale ASOS station. The above expression was added in L400-402.

Line 408-413 The QC elaborated can be applied to other meteorological sensor network whether the criteria used in the QC meet the WMO criteria for measuring temperature appropriately by correlating the purpose of the network with the phenomena to be measured. This generalization cannot be made by the method used.

ð  After pre-processing, QMS-SDM can be applied to other high-resolution meteorological network. The above description was in L425-428.

The last sentences (lines 429-431) are partially saving the paper, since they say clearly that the threshold values should be redefined to match the sensors and climatology, even in the wave cases examined.

Reviewer 2 Report

Dear Authors,

Thank you for considering my suggestions and advice. Now the paper is easy to understand. Fixed some language errors. Issues that were incomprehensible have been corrected. Now the paper can be published. Congratulations.